# Prediabetes Is Independently Associated with Subclinical Carotid Atherosclerosis: An Observational Study in a Non-Urban Mediterranean Population

**DOI:** 10.3390/jcm9072139

**Published:** 2020-07-07

**Authors:** Maria Belén Vilanova, Josep Franch-Nadal, Mireia Falguera, Josep Ramon Marsal, Sílvia Canivell, Esther Rubinat, Neus Miró, Àngels Molló, Manel Mata-Cases, Mònica Gratacòs, Esmeralda Castelblanco, Dídac Mauricio

**Affiliations:** 1Primary Health Care Center Igualada Nord, Consorci Sanitari de l’Anoia, Servei Català de la Salut, 08700 Igualada, Barcelona, Spain; mbvilanova@gmail.com; 2Biomedical Research Institute of Lleida, University of Lleida, 25198 Lleida, Spain; mireiafalguera@hotmail.com (M.F.); rubinatesther@gmail.com (E.R.); angelsmollo@gmail.com (À.M.); 3CIBER of Diabetes and Associated Metabolic Diseases (CIBERDEM), Instituto de Salud Carlos III (ISCIII), 28029 Barcelona, Spain; josep.franch@gmail.com (J.F.-N.); manelmatacases@gmail.com (M.M.-C.); 4DAP-Cat group, Unitat de Suport a la Recerca Barcelona, Fundació Institut Universitari per a la recerca a l’Atenció Primària de Salut Jordi Gol i Gurina (IDIAPJGol), 08006 Barcelona, Spain; scanivell@gmail.com (S.C.); monica.gratacos@gmail.com (M.G.); 5Department Medicine, University of Barcelona, 08036 Barcelona, Spain; 6Primary Health Care Center Raval Sud, Gerència d’Atenció Primaria, Institut Català de la Salut, 08001 Barcelona, Spain; 7Primary Health Care Center Cervera, Gerència d’Atenció Primaria, Institut Català de la Salut, Cervera, 25200 Lleida, Spain; 8Unitat de Suport a la Recerca, Institut Universitari d’Investigació en Atenció Primària Jordi Gol (IDIAP Jordi Gol), 25198 Lleida, Spain; joseprmarsal@yahoo.es; 9Epidemiology Unit of the Cardiovascular Service, Hospital Universitari Vall d’Hebron, 08035 Barcelona, Spain; 10CIBER of Epidemiology and Public Health (CIBERESP), Instituto de Salud Carlos III, 28029 Barcelona, Spain; 11Primary Health Care Center Sant Martí de Provençals, Gerència d’Atenció Primaria, Institut Català de la Salut, 0820 Barcelona, Spain; 12Department of Nursing and Physiotherapy, University of Lleida; Research Group of Health Care (GRECS), Institut de Recerca Biomèdica de Lleida, 25003 Lleida, Spain; 13Primary Health Care Center Tàrrega, Gerència d’Atenció Primaria, Institut Català de la Salut, 25300 Tàrrega, Lleida, Spain; miro.vallve@gmail.com; 14Primary Health Care Center Guissona, Gerència d’Atenció Primaria, Institut Català de la Salut, 25210 Guissona, Lleida, Spain; 15Primary Health Care Center La Mina, Gerència d’Atenció Primaria, Institut Català de la Salut, 08930 Barcelona, Spain; 16Department of Endocrinology and Nutrition, Hospital de la Santa Creu i Sant Pau and Sant Pau Biomedical Research Institute, 08041 Barcelona, Spain; 17Faculty of Medicine, University of Vic (UVic/UCC), 08500 Vic, Spain

**Keywords:** prediabetes, cardiovascular risk assessment, carotid atherosclerosis, carotid ultrasound, intima-media thickness, carotid plaques, observational study

## Abstract

This was a prospective, observational study to compare the burden of subclinical atherosclerosis as measured by carotid ultrasonography in a cohort of subjects with prediabetes vs. subjects with normal glucose tolerance (NGT) from a non-urban Mediterranean population. Atherosclerosis was assessed through carotid intima-media thickness (c-IMT), the presence/absence of carotid plaques, and plaque number. Among 550 subjects included, 224 (40.7%) had prediabetes. The mean c-IMT and the prevalence of carotid plaque were significantly higher in the prediabetes group compared to the NGT group (0.72 vs. 0.67 mm, *p* < 0.001; and 37.9% vs. 19.6%; *p* < 0.001, respectively). Older age, male gender, and increased systolic blood pressure were positively correlated with c-IMT and were independent predictors of the presence of plaques. In contrast, prediabetes and low-density lipoprotein (LDL)-c were predictors of the presence of plaque (odds ratio [OR] = 1.64; 95% confidence interval [CI] = 1.05–2.57; *p* = 0.03 and OR = 1.01; 95% CI = 1.00–1.02; *p* = 0.006, respectively) together with tobacco exposure and the leukocyte count (OR = 1.77; 95% CI = 1.08–2.89; *p* = 0.023 and OR = 1.20; 95% CI = 1.05–1.38; *p* = 0.008, respectively). In a non-urban Mediterranean population, prediabetes was associated with established subclinical carotid atherosclerosis. These findings could have implications for the prevention and treatment of CV risk in these subjects before the first symptoms of cardiovascular disease appear.

## 1. Introduction

Cardiovascular disease (CVD) is a multifactorial condition that is the result of a complex interaction between genetic predisposition and well-recognized cardiovascular (CV) risk factors such as diabetes mellitus [1]. Indeed, it is well established that patients with type 2 diabetes mellitus (T2DM) have a two- to four-fold increased risk of atherosclerotic CVD, and they have a poorer prognosis in terms of CV morbidity and mortality than subjects without diabetes [2]. Moreover, in prediabetes, which is not a clinical entity per se but an intermediate metabolic state between normoglycemia and diabetes, there is an increased risk of developing overt T2DM and CVD [3]. This is mainly because it is frequently accompanied by traditional CV risk factors such as obesity, dyslipidemia, and hypertension [4]. Some studies suggest that the development of atherosclerosis begins during the prediabetes state, long before T2DM is established and occurs concurrently with progression from normoglycemia to prediabetes and from prediabetes to overt T2DM [5]. However, since the diagnosis of prediabetes and T2DM is often delayed, macrovascular complications may already be present at the moment of the diagnosis of diabetes [6].

The thickening of the intima-media precedes the development of atherosclerotic plaques, and this interface is well depicted by ultrasound (US) imaging [7]. The measurement of carotid intima-media thickness (c-IMT) through US is a non-invasive method able to detect and quantify subclinical atherosclerosis and atherosclerotic burden in the arterial system [8]. Moreover, US can also identify the presence of carotid artery plaque, which represents later stages of atherosclerosis; carotid artery plaque is an independent predictor of CVD, and its assessment can improve the CVD risk stratification on top of the traditional risk factors [9]. Of note, carotid plaque burden is highly predictive of the development of later CV complications, and the mere presence of carotid plaque is more predictive of adverse CV outcomes than high c-IMT alone [10,11].

We have previously shown that, in subjects with T2DM without clinical CVD, the frequency and burden of subclinical atherosclerotic disease assessed through carotid US is increased compared to nondiabetic subjects [12]. Additionally, prospective US-imaging studies have consistently shown greater severity of carotid atherosclerosis in prediabetic subjects compared to non-diabetic individuals, with reports of increased c-IMT, increased prevalence of carotid plaques, and increased presence of carotid stenosis [5,13,14,15,16,17,18,19,20]. However, apart from the qualitative assessment of the carotid plaque (i.e., present or absent), the quantitative assessment of plaque burden through the number of plaques, plaque thickness, or plaque area has rarely been assessed. This is relevant because each measure may reflect different atherosclerotic phenotypes [21].

Geographical differences in urban vs. rural settings may influence the prevalence of CV risk factors that may be reflected in the prevalence of atherosclerotic plaque. For instance, the risk of long-term coronary heart disease (CHD) is higher in Northern Europe than in Mediterranean Southern Europe for the same levels of cholesterol and systolic blood pressure [22,23]. Moreover, the burden of CV risk factors is higher in rural areas than in urban areas not only in Spain but also in the rest of Europe and in the USA [24,25,26]. Conversely, the prevalence of CHD is lower in Asian populations (in particular in the rural setting) than in Caucasian populations, probably due to a lesser sensitivity to CV risk factors in terms of lifestyle and genetic factors [27].

Based on potential population and geographic differences, we aimed to prospectively assess the prevalence and burden of carotid plaques in prediabetic subjects compared to normoglycemic individuals from a non-urban area in a Mediterranean region that has not been studied previously. For this purpose, we used the Mollerussa cohort, which consists of a representative sample of healthy adults from a local county, the Pla d’Urgell in Catalonia (Spain).

## 2. Experimental Section

### 2.1. Study Population

The Mollerussa study is a prospective observational cohort study that was conducted between August 2011 and July 2015 in a non-urban area of Catalonia (Spain) among subjects from the general population. Details of the study and the protocol have been described previously [28,29]. Briefly, the study randomly selected 2226 subjects ≥25 years of age who were attended by any of the Primary Healthcare centers in the area of Pla d’Urgell (Catalonia, Spain). Subjects with a previous diagnosis of diabetes (T1, T2, or any specific subtype), treated with oral glucose-lowering drugs, or metformin for other conditions were excluded. Moreover, the study excluded subjects who had CVD (i.e., a diagnosis of heart disease, heart failure, and aortic stenosis), or who were considered to have hypertension or dyslipidemia (i.e., if they were using anti-hypertensive or lipid-lowering medications). Out of the 594 subjects initially recruited based on their willingness to participate, fulfillment of inclusion criteria, and availability of baseline glycemia measurements, the study finally assessed 583 individuals. This study included the initial cohort with available US-imaging data (*n* = 550). The study protocol was conducted following the Declaration of Helsinki and approved by the Ethics Committee of the Primary Health Care University Research Institute (IDIAP) Jordi Gol (P12/043). An informed consent form was signed for all the study participants.

### 2.2. Measures and Data Collection

As previously described, data on sociodemographic variables and anthropometric measures were recorded at baseline [28]. Moreover, laboratory parameters were obtained, and patients were classified into two groups based on glycemic status according to the fasting plasma glucose (FPG) and glycated hemoglobin (A1C) ADA criteria [30]: (1) normal glucose tolerance (NGT) group, namely subjects with FPG <100 mg/dL (<5.6 mmol/L) and/or A1C <5.7%, and (2) prediabetes group, including those with FPG between 100 mg/dL and <126 mg/dL (5.6–6.9 mmol/L) and/or A1C between 5.7% and <6.5%. Estimated glomerular filtration rate (eGFR) was calculated using the Chronic Kidney Disease Epidemiology Collaboration (CKD-EPI) formula as described previously [29].

Each study participant underwent a US study conducted by the multidisciplinary Unit for Detection and Treatment of Atherothrombotic Disease (UDETMA) from the Hospital Universitari Arnau de Vilanova (Lleida, Spain). The US was used to assess both carotids to measure c-IMT and carotid plaques following the standardized operational procedure and the Mannheim consensus [7]. This included a cross-sectional view of the common, bulb, and internal segments of both carotid arteries (CCA, BC, and ICA, respectively). In addition, atheromatous carotid plaque was defined as a focal encroachment into the arterial lumen, with online average c-IMT values in these three areas above 1.5 mm considered plaques [31].

### 2.3. Statistical Analysis

Descriptive data are presented as the mean and standard deviation (SD) for continuous outcomes, or number and percentage (%) for categorical outcomes. Student’s t-test and analysis of variance (ANOVA) were used to analyze the differences of means between the different groups. Chi-square test was used for categorical outcomes. We performed multivariate linear and logistic regression models adjusting for sex, age, tobacco exposure, low-density lipoprotein (LDL) cholesterol, high-density lipoprotein (HDL) cholesterol, triglycerides, systolic and diastolic blood pressure, waist as a measure of adiposity, urate, kidney function as eGFR and leukocytes to determine the predictors of c-IMT and carotid plaques, respectively. All significance tests were 2-tailed, and values of *p* < 0.05 were considered significant. All analyses were conducted using the Bioconductor package of the free R statistical programming language version 3.3.1 [32].

## 3. Results

Overall, 40.7% patients were diagnosed as having prediabetes (*n* = 224) and 59.3% (*n* = 326) NGT. The main characteristics of each group are shown in Table 1. There were no significant differences in the proportion of men/women between groups, but subjects with prediabetes were older and had a poorer CV risk profile in terms of adiposity (*p* < 0.001), dyslipidemia and hypertension (*p* = 0.002), tobacco exposure and kidney function (*p* < 0.001). In addition, the leukocyte count was higher in the prediabetic group (*p* = 0.033).

### 3.1. Carotid-IMT and Plaque Burden

The mean c-IMT in the total cohort was 0.69 mm (SD = 0.1), and it was significantly higher in the prediabetes group compared to the NGT group (0.72 vs. 0.67 mm, *p* < 0.001, Table 2). The overall prevalence of atherosclerotic carotid plaque was 27.1%, and it was present in a higher proportion of subjects with prediabetes than in the NGT group (37.9% vs. 19.6%, *p* < 0.001). In the overall population, 14% of individuals had one plaque, while 13.1% had multiple plaques (Table 2). This percentage also varied according to the glycemic status, with a higher proportion of subjects with one or multiple plaques in the prediabetes group (19.6% and 18.3%, respectively) than in the NGT group (10.1% and 9.5%, respectively, *p* < 0.001, Table 2).

### 3.2. Predictors of c-IMT and Atherosclerotic Plaque Burden

The linear regression analysis for the predictors of c-IMT revealed that the only variables associated with increased thickness were male gender (*p* < 0.001), older age (*p* < 0.001), and elevated systolic blood pressure (*p* = 0.007), but not prediabetes (Table 3). The logistic regression analysis (after controlling for conventional atherosclerotic risk factors) showed that the presence of any carotid plaque was associated with prediabetes (OR = 1.64, *p* = 0.034), male gender (OR = 1.94, *p* = 0.026), older age (OR = 1.08, *p* < 0.001), tobacco exposure (OR = 1.70, *p* = 0.036), LDL-cholesterol (OR = 1.01, *p* = 0.013), and leukocyte count (OR = 1.18, *p* = 0.022) (Table 4). Further information on the univariate analysis of carotid plaque presence is shown in Appendix A.

## 4. Discussion

Our study reveals that subjects with prediabetes from a non-urban population had an increased c-IMT and prevalence and burden of carotid plaques compared to NGT subjects. However, although prediabetes was not correlated with c-IMT, this entity was an independent predictor of plaque burden. Considering that carotid plaque is in turn an independent predictor of CVD, our data suggest that, in our study population, almost 38% of patients with prediabetes are at a very high risk of developing CVD.

The mean c-IMT difference between subjects with prediabetes and subjects with normoglycemia reported by a meta-analysis of 13 studies using US imaging was 0.04 mm (SD, 0.012–0.048) [13]. This figure has been confirmed in further studies, and it is also in line with the difference observed in our study (0.05 mm) [14,15,16,18]. Regarding carotid plaques, our research found that the prevalence was 38% among subjects with IGT and 20% in the NGT group. These figures agree, albeit in a low range, with the prevalence reported by similar US imaging studies that incorporated carotid plaque assessment in addition to c-IMT, ranging between 25–45% in IGT subjects and between 21–34% in NGT subjects [15,16,17,33].

The frequency of CV risk factors varies between populations and geographical areas [22,23,24,25,26,27]. In our study, the first to be conducted in a Mediterranean non-urban area, the prevalence of carotid plaque in prediabetes was lower than that observed in two other studies conducted in an urban area or in general population of Southern European regions (38% vs. 45–49%) [16,20]. In contrast, the reported frequency in urban areas of Japan and China was higher than that in our study (24–28% vs. 38%) [5,15,17,33,34]. These differences match previous epidemiologic studies of a lower CV risk profile in Asian populations, but do not agree with the higher frequency reported in rural vs. urban areas in Caucasian populations. This apparent discrepancy could be explained by methodological differences between studies, including the characteristics of the studied population and/or of the carotid US measures assessed. Regarding the study population, we included subjects younger than those of previous similar studies (mean age 50 years in our study vs. 57–68 years in others), which may be clearly associated with a lower CV risk in our case. Moreover, and most importantly, our study was conducted on subjects from the general population a priori free of CVD, while reports assessing individuals with CV risk factors or established CVD observed a much higher prevalence of carotid plaques in both control and prediabetic groups (44–53% and 49–79%, respectively) [5,17,20,34]. Regarding differences in US measurements, the definition of carotid plaque frequently varies between studies. For instance, the use of less strict criteria to consider carotid plaque might have resulted in lower sensitivity for the diagnosis of carotid plaque, which could have led to a higher number of subjects being classified as plaque-free in some studies [15,33]. Overall, and despite methodological differences, independent US imaging studies confirm our results that carotid plaque burden (both presence and number of plaques) is increased among subjects with prediabetes compared with those with normoglycemia. This is also in agreement with studies conducted using magnetic resonance (MRI), which have found a prevalence between 16–18% in control subjects and 33–35% in subjects with prediabetes [35,36].

In our study, older age, male gender, and increased systolic blood pressure were positively correlated with c-IMT and older age and male gender were independent predictors of plaque presence. Additionally, prediabetes and LDL-cholesterol levels were not correlated with c-IMT, but they were predictors of the presence of plaque together with tobacco exposure. The same variables have been previously shown as predictors of c-IMT and the same ones as predictors of the presence of carotid plaque in different populations [37,38], including those assessing subjects with prediabetes [5,16,18,19,20]. These results may reflect the fact that, although c-IMT and plaque are correlated processes, they capture the impact of different factors; in actuality, c-IMT may reverse to normal values as a result of improvements in some factors. Indeed, while age and hypertension are primary contributors to c-IMT, thickening, age and dyslipidemia are the main predictors of carotid plaque [21,39]. Adding to these differences, our study showed that increased leukocytes independently predicted the presence of plaque but was not related to increased c-IMT. Atherosclerosis is a chronic inflammatory disease, and plaque a localized manifestation of it where leukocyte recruitment, oxidation, endothelial dysfunction, and/or smooth cell proliferation are involved [40]. Indeed, epidemiological studies have shown a positive correlation between white blood cell (WBC) count and the risk of CVD in subjects with or without CVD [41]. Moreover, WBC count is positively correlated with A1C levels, and elevated count has been associated with carotid and aortic plaque thickness in subclinical atherosclerosis [42,43,44]. Finally, low-grade chronic inflammation is a key component of T2DM, where an increased WBC count has been reported as independently associated with increased incidence and risk of the disease [45,46]. Of note, WBC counts have also been reported to be higher in subjects with IGT than in normoglycemic subjects [47,48], as also observed in our study. It is therefore possible that elevated leukocyte count in subjects with prediabetes may be an early marker of disease and of initiation or progression of significant atherosclerotic plaques, in turn leading to an increased risk of CVD.

Although it is beyond dispute that subjects with prediabetes are at increased risk for CVD, it remains unclear whether screening for subclinical atherosclerosis in this subpopulation is appropriate. The confirmation that the proportion of carotid plaques is higher among these individuals than in normoglycemic subjects is relevant in daily clinical practice for several reasons. Firstly, because some clinical trials on new antidiabetic treatments (e.g., dipeptidyl peptidase-4 inhibitors; DPP-4) have shown to attenuate c-IMT progression compared with conventional treatment in subjects with T2DM and no CVD [21]. Secondly, because of the evidence that CV mortality in people with IGT is statistically lower if they undergo lifestyle intervention instead of being treated as per standard of care [49]. Considering all of the evidence, it would seem appropriate to identify subjects with prediabetes with associated subclinical atherosclerosis in daily clinical practice to enforce more stringent CV prevention strategies for them.

This study has some strengths and limitations worth mentioning. The main strength is the prospective, population-based design, which reflects real-world data from a non-urban area not previously studied. Moreover, it was specifically designed to detect prediabetes and, besides the careful evaluation of traditional CV risk factors, the US carotid imaging was based on standardized clinical procedures, conducted by trained staff, and included the plaque burden in the c-IMT measurements. One limitation of the study is that the FPG and A1C tests were performed only once, and we did not conduct the oral glucose tolerance test (OGTT) as a criterion for prediabetes definition. The latter is because OGTT is less accessible than FPG or A1C in clinical practice, has low reproducibility, and is inconvenient in terms of costs and time consumption [50]. However, while all three glycemic indices (A1C, FPG, and OGTT) have been shown to predict c-IMT, only FPG seems to predict the presence of carotid plaques [15]. Another limitation is that the design of the Mollerussa Cohort Study was focused on the evaluation of different primary endpoints, i.e., the prevalence of prediabetes and undiagnosed diabetes in that particular population. Furthermore, the current study may not have sufficient power to detect the contribution of some risk factors, such as LDL, to c-IMT thickening. In addition, prospective studies are needed to properly address the contribution of all CV risk factors to the progression of atherosclerotic disease.

## 5. Conclusions

The results of this observational study support that, in subjects from a non-urban Mediterranean area, prediabetes predicts the presence of atherosclerotic plaques as measured by US-imaging. This is relevant for identifying individuals at a high risk of subclinical atherosclerotic disease, who could benefit from intensive preventive treatment before clinical CVD appears.

## Figures and Tables

**Table 1 jcm-09-02139-t001:** Sociodemographic and clinical characteristics of the subjects included in the study.

	All Subjects	NGT	Prediabetes	*p*-Value
Sample size, N (%)	550	326 (59.3%)	224 (40.7%)	
Gender (women), n (%)	321 (58.4)	190 (58.3)	131 (58.5)	0.963
Age (years), mean (SD)	50.1 (13)	47.3 (12.8)	54.3 (12.2)	<0.001
BMI (kg/m^2^), mean (SD)	26.1 (4.5)	25.3 (4.3)	27.3 (4.5)	<0.001
Waist (cm), mean (SD)	93.8 (12.2)	91.8 (12.0)	96.7 (12.0)	<0.001
Tobacco exposure, n (%)	283 (51.5%)	162 (49.7%)	121 (54%)	<0.001
FPG (mg/dL), mean (SD)	90.8 (10.3)	86.6 (7.0)	97 (11.2)	<0.001
A1C (%), mean (SD)	5.5 (0.4)	5.2 (0.3)	5.8 (0.3)	<0.001
Dyslipidemia, n (%)	65 (11.8%)	27 (8.3%)	38 (17%)	0.002
Total-c (mg/dL), mean (SD)	201 (36.2)	197.8 (38.3)	205.7 (32.4)	0.006
HDL-c (mg/dL), mean (SD)	58.7 (14.7)	58.7 (14.8)	58.8 (14.4)	0.911
LDL-c (mg/dL), mean (SD)	122 (30.7)	119.4 (31.3)	125.6 (29.4)	0.010
Triglycerides (mg/dL), mean (SD)	107.2 (80.9)	104.5 (90.8)	111 (63.7)	0.070
Hypertension (%), n (%)	84 (15.3)	37 (11.3)	47 (21)	0.002
SBP (mmHg), mean (SD)	122 (16.7)	119.2 (16.3)	125.9 (16.5)	<0.001
DBP (mmHg), mean (SD)	76.8 (10.1)	75.7 (10.1)	78.4 (9.9)	0.001
Creatinine (mg/dL), mean (SD)	0.8 (0.2)	0.8 (0.2)	0.8 (0.2)	0.412
eGFR (ml/min), mean (SD)	94.2 (15.1)	96.7 (14.1)	90.6 (15.8)	<0.001
Serum urate (mg/dL), mean (SD)	4.9 (1.3)	4.8 (1.2)	5 (1.3)	0.056
ALT (U/L), mean (SD)	20.4 (17.4)	20.4 (20.5)	20.3 (11.5)	0.190
Leukocytes (× 10^9^/L), mean (SD)	6.6 (1.7)	6.4 (1.6)	6.8 (1.8)	0.033

A1C, glycated hemoglobin; ALT, alanine transaminase; BMI, body mass index; DBP, diastolic blood pressure; eGFR, estimated glomerular filtration rate; FPG, fasting plasma glucose; HDL-c, high-density lipoprotein cholesterol; LDL-c, low-density lipoprotein cholesterol; NGT, normal glucose tolerance; SBP, systolic blood pressure; SD, standard deviation; Total-c, total cholesterol.

**Table 2 jcm-09-02139-t002:** Carotid intima-media thickness (c-IMT) and plaque burden among patients with prediabetes or normal glucose tolerance.

	All Subjects	NGT	Prediabetes	*p*-Value
c-IMT, mm, mean (SD)	0.69 (0,1)	0.67 (0.1)	0.72 (0.1)	<0.001
Presence of carotid plaque, n (%)				
No plaque	401 (72.9)	262 (80.4)	139 (62.1)	<0.001
Significant plaque	149 (27.1)	64 (19.6)	85 (37.9)	<0.001
*Subjects with 1 plaque*	77 (14)	33 (10.1)	44 (19.6)	<0.001
*Subjects with multiple plaques*	72 (13.1)	31 (9.5)	41 (18.3)	<0.001
Number of carotid plaques				
Mean (SD)	0.47 (0.9)	0.36 (0.9)	0.64 (1.0)	<0.001
Median (IQR)	0 (0–1)	0 (0–0)	0 (0–1)	

c-IMT, carotid intima-media thickness; IQR, interquartile range; NGT, normal glucose tolerance; SD, standard deviation.

**Table 3 jcm-09-02139-t003:** Linear regression analysis for the predictors of c-IMT.

	Coefficient	95% CI	*p*-Value
Prediabetes	0.47	−1.23–2.17	0.589
Gender, male	3.99	1.85–6.14	<0.001
Age	0.47	0.38–0.56	<0.001
Tobacco exposure	1.31	−0.38–3.00	0.128
Waist	0.01	−0.07–0.10	0.738
LDL cholesterol	0.03	−0.00–0.05	0.067
HDL cholesterol	−0.02	−0.08–0.05	0.645
Triglycerides	0.00	−0.01–0.01	0.976
eGFR	−0.04	−0.11–0.03	0.244
Leukocytes	−0.11	−0.62–0.41	0.683
Serum urate	−0.13	−1.01–0.75	0.770
Systolic blood pressure	0.11	0.03–0.19	0.007
Diastolic blood pressure	−0.08	−0.20–0.04	0.206

CI, confidence interval; LDL, low-density lipoprotein; HDL, high-density lipoprotein; eGFR, estimate glomerular filtration rate.

**Table 4 jcm-09-02139-t004:** Logistic regression analysis for the predictors of carotid plaques.

	OR	95% CI	*p*-Value
Prediabetes	1.64	1.04–2.58	0.034
Gender, male	1.94	1.09–3.49	0.026
Age	1.08	1.06–1.11	<0.001
Tobacco exposure	1.70	1.04–2.81	0.036
Waist	1.02	0.99–1.04	0.216
LDL cholesterol	1.01	1.00–1.02	0.013
HDL cholesterol	1.00	0.98–1.02	0.969
Triglycerides	1.00	0.10–1.00	0.580
eGFR	1.01	0.99–1.03	0.574
Leukocytes	1.18	1.03–1.37	0.022
Serum urate	0.95	0.75–1.21	0.696
Systolic blood pressure	1.02	0.99–1.04	0.158
Diastolic blood pressure	0.10	0.96–1.03	0.811

OR, odds ratio; CI, confidence interval; LDL, low-density lipoprotein; HDL, high-density lipoprotein; eGFR, estimated glomerular filtration rate.

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
