# Peer review of "Prediabetes Is Independently Associated with Subclinical Carotid Atherosclerosis: An Observational Study in a Non-Urban Mediterranean Population"

_jcm, 2020, doi:10.3390/jcm9072139_

Round 1
Reviewer 1 Report
I read with interest the work of MB Vilanova and coworkers reporting the analysis of factors associated with c-IMT and carotid plaque, in a prospective observational study conducted in a non-urban Mediterranean population.
The study was conducted in adult patients characterized by low cardiovascular risk. In a first step they analyzed differential characteristics associated with NGT and Prediabetes groups, defined according to FGP and A1C. As expected, patients with prediabetes were older, had higher BMI, were more frequently smokers, had higher cholesterolemia, more frequently hypertension and had lower eGFR. Patients with prediabetes had higher c-IMT and more frequently carotid plaque.
In a next step, they analysed factors associated with c-IMT and atherosclerotic plaques. They first analyzed factors associated with c-IMT and found male gender, older age and elevated blood pressure but not prediabetes to be associated with c-IMT. Logistic regression for prediction of carotid plaque revealed prediabetes, older age, tobacco, high LDL, high blood pressure and leucocyte count to be associated with plaques.
The work is mainly confirmative in nature as such observations have already been described in other studies, as underlined in the introduction by the authors. However, the originality of the work relies on the fact that the study was conducted in a non-urban population, showing that prediabetes is associated with carotid plaques, allowing to design preventive strategies.
The article is well written and conclusions are supported by data.
The strength of the work is its prospective design. However, a major limitation is that NGT/Prediabetes classication relies in an only measurement, thus some subjects may have been misclassified or classified as begin prediabetic instead of diabetic.
I have some questions and some suggestions:
- Table 3:
- The authors analyzed in a linear regression analysis factors associated with c-IMT. In the table, female gender should be replaced by male gender, as the authors refer to male gender in manuscript.
- The description is limited to significant associated factors. The association with other factors should be included in the table.
- Figure 1:
- the figure should be replaced by a table.
- The authors show only significant associations. Is there any association between carotid plaques and kidney function?
- Is it a multivariate regression analysis? If yes, I also suggest to show univariate analysis and then multivariate analysis.
- In the discussion, the authors say that despite c-IMT and carotid plaque are pathophysiologically linked, they “capture impact” of different factors. As underlined by authors previous studies have identified common risk factors. Thus, their observation may only been explained by the size of the study. In other words, maybe the study has not the power to identify prediabetes and LDL-c as risk factors for c-IMT.
- Please provide research authorization agreement and ethical form.

Author Response
Prediabetes is Independently Associated with Subclinical Carotid Atherosclerosis: an Observational Study in a Non-urban Mediterranean Population (manuscript reference jcm-835809).
We appreciate the input given by the Reviewers, which enabled us to greatly improve the quality of our manuscript. In the following pages, we enclose our point-by-point responses to each of the Reviewer comments. Please, note that in the revised version of the Manuscript, changes are highlighted in track changes function so that they can be easily traced.
REVIEWER #1
Table 3:
- The authors analyzed in a linear regression analysis factors associated with c-IMT. In the table, female gender should be replaced by male gender, as the authors refer to male gender in manuscript.
We thank the Reviewer for this observation. In the new version of the manuscript, we have amended the data to consider women as the reference category (please, see Results section; Table 3; page 5).
- The description is limited to significant associated factors. The association with other factors should be included in the table.
As suggested by the Reviewer, we have included all potential risk factors in the model shown in Table 3, including kidney function. Please, also note that we included waist as the variable used to measure adiposity. Further, the Experimental Section has been modified to clarify the details of all the variables included in the multivariate lineal and logistic regression models.
Please, see Table 3 in Results section; Page 5
We have further explained this in the Experimental Section; Statistical Analysis; Page 4 (Lines 152 – 156)
“We performed multivariate linear and logistic regression models adjusting for sex, age, tobacco exposure, LDL cholesterol, HDL cholesterol, triglycerides, systolic and diastolic blood pressure, waist as a measure of adiposity, urate, kidney function as eGFR and leukocytes to determine the predictors of c-IMT and carotid plaques, respectively.”
Figure 1:
- The figure should be replaced by a table.
Following the Reviewer’s advice, we have now replaced Figure 1 by the new Table 4 in the Results section, Page 6.
- The authors show only significant associations. Is there any association between carotid plaques and kidney function?
We also thank the Reviewer for this comment. As shown in the previous Reviewer’s question, we have replaced Figure 1 by the new Table 4, which contains additional risk factors (i.e., diastolic blood pressure, urate, waist, triglycerides, and HDL cholesterol), and we have also included kidney function. Moreover, and as specified in our response to Question #2 regarding Table 3, the Experimental section has been modified to include details on the variables included in the multivariate lineal and logistic regression models.
- Is it a multivariate regression analysis? If yes, I also suggest to show univariate analysis and then multivariate analysis.
Following the Reviewer’s suggestion, we added the univariate analysis in the new version of the manuscript as Table 1 within the Supplementary Material. Moreover, we have added a sentence in the Results section to refer the reader to this new Table (Section 3.2.; Page 5; Lines 194 – 195):
“Further information on the univariate analysis of carotid plaque presence is shown in Supplementary Table 1.”
In the discussion, the authors say that despite c-IMT and carotid plaque are pathophysiologically linked, they “capture impact” of different factors. As underlined by authors previous studies have identified common risk factors. Thus, their observation may only been explained by the size of the study. In other words, maybe the study has not the power to identify prediabetes and LDL-c as risk factors for c-IMT.
We appreciate the Reviewer’s comment on the possibility that the current study is not powered to identify prediabetes and LDL-c as risk factors for c-IMT. In the new version of the manuscript, we have added a piece of text in the paragraph on limitations in the Discussion section. (Section 4.; Page 8; Lines 290 – 293).
“Further, the current study may not have sufficient power to detect the contribution of some risk factors, such as LDL, to c-IMT thickening. In addition, prospective studies are needed to properly address the contribution of all CV risk factors to the progression of atherosclerotic disease.”
Please provide research authorization agreement and ethical form.
We appreciate the Reviewer’s comment, and we apologize for this omission. In the new version of the manuscript, we are including this information (Experimental section; Study population design subsection; Page 3; Lines 128 – 130):
“The study protocol was conducted following the Declaration of Helsinki and approved by the Ethics Committee of the Primary Health Care University Research Institute (IDIAP) Jordi Gol (P12/043). An informed consent form was signed for all the study participants.”

Reviewer 2 Report
This was a prospective, observational study to compare the burden of subclinical atherosclerosis as measured by carotid ultrasonography in a cohort of subjects with prediabetes vs. subjects with normal glucose tolerance (NGT) from a non-urban Mediterranean population. Atherosclerosis was assessed through carotid intima-media thickness (c-IMT), the presence/absence of carotid plaques, and plaque number. Among 550 subjects included, 224 (40.7%) had prediabetes. The mean c-IMT and the prevalence of carotid plaque were significantly higher in the prediabetes group compared to the NGT group (0.72 vs. 0.67 mm, p<0.001; and 37.9% vs. 19.6%; p<0.001, respectively). Older age, male gender, and increased systolic blood pressure were positively correlated with c-IMT and were independent predictors of the presence of plaques. In contrast, prediabetes and LDL-c were predictors of the presence of plaque (odds ratio [OR]=1.64; 95% confidence interval [CI]= 1.05-2.57; p=0.03 and OR=1.01; 95% CI=1.00-1.02; p=0.006, respectively) together with tobacco exposure and the leukocyte count (OR=1.77; 95% CI=1.08-2.89; p=0.023 and OR=1.20; 95% CI=1.05-1.38; p=0.008, respectively). In a non-urban Mediterranean population, prediabetes was associated with established subclinical carotid atherosclerosis. Hyperglycemia could have implications for the prevention and treatment of the CV risk in these subjects before the first symptoms of cardiovascular disease appear.
The study is well-done and the paper is written very well. The novelty is examination in a non-urban population which is important as exposure to other cardiovascular risk factors, as particulate air pollution, may be less than in urban populations. Nevertheless, the conclusion is similar to previous studies in urban settings - that prediabetes is associated with subclinical carotid atherosclerosis and therefore, these subjects should have aggressive treatment of CV risk factors.
Author Response
Prediabetes is Independently Associated with Subclinical Carotid Atherosclerosis: an Observational Study in a Non-urban Mediterranean Population (manuscript reference jcm-835809).
We appreciate the input given by the Reviewers, which enabled us to greatly improve the quality of our manuscript. In the following pages, we enclose our point-by-point responses to each of the Reviewer comments. Please, note that in the revised version of the Manuscript, changes are highlighted in track changes function so that they can be easily traced.
REVIEWER #2
The study is well-done and the paper is written very well. The novelty is examination in a non-urban population which is important as exposure to other cardiovascular risk factors, as particulate air pollution, may be less than in urban populations. Nevertheless, the conclusion is similar to previous studies in urban settings - that prediabetes is associated with subclinical carotid atherosclerosis and therefore, these subjects should have aggressive treatment of CV risk factors.
We highly appreciate the positive view of this Reviewer. We understand that, apart from the changes made on request from Reviewer 2, no further changes are needed.
